# IgE-Mediated Food Sensitization, Management Strategies, and Quality of Life in Pediatric Eosinophilic Esophagitis: A Prospective Observational Study

**DOI:** 10.3390/nu17121980

**Published:** 2025-06-11

**Authors:** Lisa Nuyttens, Toon Dominicus, Cheyenne Keppens, Tine Alliet, Sophie Verelst, Marianne Diels, Tessa Bosmans, Rik Schrijvers, Ilse Hoffman, Dominique M. A. Bullens

**Affiliations:** 1Allergy and Immunology Research Group, Department of Microbiology, Immunology and Transplantation, KU Leuven, 3000 Leuven, Belgium; cheyenne.keppens@kuleuven.be (C.K.); sophie.verelst@uzleuven.be (S.V.); rik.schrijvers@kuleuven.be (R.S.); dominique.bullens@kuleuven.be (D.M.A.B.); 2Clinical Division of Pediatrics, University Hospitals Leuven, 3000 Leuven, Belgium; toon.dominicus@uzleuven.be (T.D.); tine.alliet@uzleuven.be (T.A.); marianne.diels@uzleuven.be (M.D.); tessa.bosmans@uzleuven.be (T.B.); 3Division of General Internal Medicine, Allergy and Clinical Immunology, UZ Leuven, 3000 Leuven, Belgium; 4Pediatric Gastroenterology, Hepatology and Nutrition, University Hospitals Leuven, 3000 Leuven, Belgium; ilse.hoffman@uzleuven.be

**Keywords:** pediatric eosinophilic esophagitis, food sensitization, QoL

## Abstract

**Background:** Eosinophilic esophagitis is a chronic immune-mediated disease with increasing prevalence. In pediatric populations, it often coexists with IgE-mediated food sensitization. This dual diagnosis presents unique therapeutic challenges, including on the one hand both temporary and lifelong dietary restrictions, and on the other hand, therapeutic interventions with a potential impact on quality of life (QoL). **Objectives**: This study prospectively evaluated the prevalence of IgE-mediated food sensitization and allergy in children with EoE attending a tertiary center in Flanders, Belgium. Additionally, it prospectively documented dietary habits and restrictions in these children, with or without concomitant IgE-mediated food allergies, and assessed the impact of dietary management on quality of life compared to pharmacological treatment. **Methods**: We consecutively followed 30 children with biopsy-confirmed pediatric EoE (PedEoE) attending a tertiary referral center for at least 12 months. Patient demographics, allergy testing, dietary history, and treatment modalities were recorded. Symptom score and PedEoE QoL were assessed using validated Pediatric Eosinophilic Esophagitis Symptom Score (PEESS 2.0) and Pediatric Quality of Life Inventory (PedsQL 3.0) questionnaires. Statistical analysis was performed using the Mann–Whitney U test and Kruskal–Wallis test with Dunn’s test as a post hoc test. **Results**: Among 30 children with EoE (60% male, median age 8 years), 60 PedEoE QoL (PedsQL) and 39 symptom (PEESS) questionnaires were collected at one or more time points over a median follow-up of 14.5 months. IgE sensitization to common dietary triggers was observed in multiple patients, with varying clinical reactivity. Symptom scores tended to be higher in children without histological remission, though differences were not statistically significant (median PEESS 23.75 vs. 17.50, *p* = 0.1934). Grouped by degree of dietary restriction, QoL scores showed significant differences (child *p* = 0.0102; parent *p* = 0.0203), with children in the 1–2 food elimination group reporting better QoL compared to the 0 food elimination and >6 food elimination groups. No clear statistically significant differences were observed between the 1–2 food elimination group and the 3–4 or 5–6 food elimination groups. **Conclusions**: IgE sensitization is prevalent among pediatric EoE patients and has significant clinical implications for disease management. Treatment choice can influence not only disease control but also the QoL of both patients and their families.

## 1. Introduction

Eosinophilic esophagitis (EoE) is a chronic food antigen-mediated disease characterized by a local eosinophilic inflammation of the esophagus. It is estimated to affect approximately 1 in 2000 individuals, but its incidence has been increasing. In pediatric EoE (PedEoE), symptoms vary depending on the age of the child, with vomiting, nausea, and feeding difficulties resulting in failure to thrive being prominent in young children and night-time cough, epigastric pain, dysphagia, and even food impaction in older children and adolescents [1,2,3]. The diagnosis of EoE requires a combination of symptoms of esophageal dysfunction, macroscopic evidence of esophageal inflammation observed during endoscopy, and histological confirmation of eosinophilic infiltration—defined as at least 15 eosinophils per high-power field—in esophageal biopsy specimens obtained from multiple levels of the esophagus [1,4]. The management of PedEoE necessitates a multidisciplinary approach. According to the PedEoE guidelines published in 2014, high-dose proton pump inhibitors (PPIs) were recommended as the first-line treatment. For patients who did not achieve remission with PPI therapy, the guidelines recommended either topical corticosteroid (tCS) therapy or dietary elimination strategies [5]. The updated PedEoE guidelines now recommend PPIs, tCS, and dietary therapy as equally valuable first-line treatments for PedEoE [4]. The choice among these options should be tailored to the individual patient’s clinical situation and preferences. In Europe, the six most common dietary triggers for EoE are cow’s milk (CM), hen’s egg (HE), wheat, soy, (pea)nuts, and (shell)fish. When dietary elimination is selected, it involves the complete avoidance of one or more of these triggers, implemented either through a step-down or step-up approach [6]. Therapeutic efficacy is evaluated using a combination of symptom scoring and follow-up endoscopies. The primary objectives of treatment include achieving symptom relief, improving quality of life (QoL), and preventing long-term complications such as esophageal strictures and fibrosis, which are associated with persistent eosinophilic infiltration [7].

Although EoE is generally considered a non-IgE-mediated disorder in adults, more recent literature indicates that 40–74% of children diagnosed with EoE exhibit concomitant IgE-mediated food sensitization or allergy [1,8]. Additionally, the occurrence of EoE following tolerance induction protocols or during oral immunotherapy for a specific food further suggests a link between EoE and IgE-mediated food allergy [8]. This highlights the necessity of tailoring therapeutic approaches to each individual patient. A standardized dietary strategy may not be universally applicable in patients with EoE, particularly in the context of concomitant IgE-mediated food allergies. For instance, the reintroduction of a specific food in a patient with an IgE-mediated allergy after tolerance acquisition over time may—even if perfectly tolerated—still lead to EoE symptoms if that food is also a trigger of their EoE. This underscores the importance of a personalized approach that considers both the immunological complexity and the potential interplay between EoE and IgE-mediated food allergy.

The primary aim of this study is to prospectively document dietary habits and restrictions over a period of at least 12 months in children diagnosed with EoE, with or without concomitant IgE-mediated food allergy, at a single center. A secondary aim is to evaluate the impact of dietary management—compared to pharmacological treatments (PPI or tCS)—on the QoL of both patients and their families. The study also explores whether adjustments in dietary guidance affect QoL outcomes in this population.

## 2. Materials and Methods

### 2.1. Study Population

We consecutively enrolled 30 children (age ≤ 16 years at baseline) with PedEoE, attending the pediatric gastroenterology or pediatric allergology department at a tertiary center between October 2020 and November 2023, and in whom diagnosis was established between 1 January 2014 and 31 December 2021. Diagnosis was based on symptoms and histological findings of ≥15 eosinophils per high-power field on esophageal biopsy [4]. Based on data on EoE patient volumes at our institution, we employed a convenience sampling approach, enrolling consecutive eligible patients during the recruitment period. This method enabled the completion of patient recruitment within the desired timeframe and ensured that all participants had at least one year of follow-up data. Exclusion criteria for the study were age above 16 years at the time of inclusion and other conditions that cause esophageal eosinophilia, such as gastro-esophageal reflux disease. Clinical data were collected from children diagnosed with EoE at our tertiary center. Patient characteristics, including age, sex, age at diagnosis, and relevant medical history, were extracted from electronic medical records. In addition, data on allergy testing (including specific IgE levels), histological findings (e.g., eosinophil counts per high-power field), treatment regimens (dietary and/or pharmacological), and disease course were systematically recorded. Initially, our institutional protocol recommended a stepwise approach, beginning with PPIs, followed by dietary interventions or tCS if no remission occurred. However, based on current guidelines, we now advocate for a more individualized treatment strategy, with PPI therapy, dietary interventions, and tCS considered equally viable first-line options. The choice among these therapies is made collaboratively with the patient and their family, taking into account pre-existing dietary habits and any ongoing treatments. Information was retrieved during routine outpatient visits and supplemented with findings from endoscopic procedures and allergy consultations. Data collection was performed at baseline (time of inclusion) and at each subsequent follow-up consultation, according to the clinical follow-up schedule. Parents provided written informed consent, and children provided assent when appropriate. These 30 children with PedEoE were prospectively followed for at least 12 months at the pediatric gastroenterology or allergology division at the University Hospitals Leuven, a pediatric tertiary referral center in Belgium. At different standard of care visits, these children and their parents were asked to complete two validated Dutch-translated questionnaires: the Pediatric Quality of Life Inventory (PedsQL, version 3.0) and the Pediatric Eosinophilic Esophagitis Symptom Score (PEESS, version 2.0) [9,10]. PedsQL encompasses 7 Scales: Symptoms I (consisting of 6 items), Symptoms II (4 items), Treatment (5 items), Worry (6 items), Communication (5 items), Food and Eating (4 items), and Food Feelings (3 items). Each of these items recalls the past week. Items are reverse-scored and linearly transformed to a 0–100 scale. Child self-report forms and parent proxy-report forms in parallel are specific for ages 5–7 (young child), 8–12 (child), and 13–18 years (adolescent). In addition, there is also a parent proxy-report form for children aged 2–4 years (toddlers). PEESS is a self-report patient-reported outcome measure for children 8–18 years designed to assess symptom severity and frequency of the past month in children with EoE. The PEESS questionnaire includes items covering various symptom domains, such as dysphagia, pain, and food-related difficulties, as reported by patients. Items are linearly transformed to a 0–100 scale.

### 2.2. Data Analysis

Data analysis was performed using GraphPad Prism (version 9.3.0 (463) for Windows, GraphPad Software, San Diego, CA, USA). Continuous variables were reported as medians with interquartile ranges (IQR). For the analysis of patient-reported outcomes (QoL and PEESS), both repeated measures (i.e., all available questionnaires per child across follow-up) and cross-sectional data (i.e., only the first completed questionnaire per patient) were used. In Section 3, we clearly indicate for each analysis whether it was based on the full dataset of completed questionnaires or limited to the first time point per child. QoL assessments were conducted using the Mann–Whitney U test for comparisons between two groups, while the Kruskal–Wallis test, followed by Dunn’s post hoc test, was applied for comparisons involving more than two groups. Statistical significance was established as *p* < 0.05. Specific IgE levels were measured using the CAP system (Thermo Fisher Scientific, Waltham, MA, USA) with a sensitization cut-off of > 0.10 kU/L. The presence of sIgE at these lower levels does not always correlate with clinical allergy. However, a threshold of >0.10 kU/L was applied in our study to capture all potentially relevant sensitizations, while recognizing that clinical correlation remains essential [11].

### 2.3. Ethical Approval

Ethical approval was obtained from the Institutional Review Board. Patient confidentiality was maintained throughout the study.

## 3. Results

### 3.1. Demographics

The cohort of 30 PedEoE patients comprised 18 boys (60%) and 12 girls (40%). The median age at EoE diagnosis was 8 years (IQR 2.9–11.50), while the median age at study inclusion was 11 years (IQR 5.7–15.0). The median duration of follow-up after inclusion was 14.5 months (IQR 10.5–21.0). The medical history of most included patients was unremarkable. However, two patients (Patient 26 and Patient 30) were born with esophageal atresia with tracheoesophageal fistula and underwent surgical correction in the first days of life. Patient (P) 21 had duodenal atresia with congenital esophageal stenosis requiring multiple dilations. Additionally, P12 was diagnosed with cystic fibrosis. During study follow-up, the number of gastroscopies per child ranged from zero to seven: 8 children had none, 13 children had one, 3 children had two, 3 children had three, 1 child had four, 1 child had five, and 1 child underwent seven gastroscopies. In some cases, follow-up endoscopy was not performed due to clinical remission and shared decision-making between clinicians and caregivers, or because the child or parents declined the procedure.

### 3.2. IgE-Mediated Food Allergy in EoE

The six most common associated dietary triggers for EoE are CM, HE, wheat, soy, (pea)nuts, and (shell)fish. Coincidentally, these six are also the statistically most likely foods to cause an IgE-mediated food allergy in children.

#### 3.2.1. IgE Sensitization to Cow’s Milk

In our PedEoE cohort, seven out of thirty children had detectable sIgE to CM (Figure 1, Table 1). Clear clinical reactions were observed in two of these seven children. P2 developed widespread urticaria along with localized facial swelling and pruritus, whereas P14 experienced diarrhea following CM ingestion. The other children with elevated sIgE to CM experienced mostly mild atopic dermatitis, for which CM products were excluded, with a positive effect on the skin symptoms. During follow-up, P11 successfully introduced highly heated CM (sIgE to cow’s milk and components ≤ 0.24 kU/L over time). However, symptoms of dysphagia emerged after four weeks, leading to the discontinuation of heated CM ingestion. Recurrence of EoE was confirmed by biopsy four weeks later. P14 first underwent an oral food challenge (OFC) with highly heated (20 min cooked) CM based on decreasing sIgE levels to cow’s milk (4.86 kU/L to 0.56 kU/L) and negative IgE to casein. The introduction of highly heated CM was well tolerated without immediate reactions or EoE reactivation. At study completion (3 months after the introduction of highly heated CM), P14 underwent an additional OFC with uncooked CM and again tolerated it without adverse effects or subsequent EoE reactivation on gastroscopy.

#### 3.2.2. IgE Sensitization to Hen’s Egg

In our cohort of thirty children, seven children had elevated IgE levels to HE (Figure 1, Table 2), with three of these children exhibiting clinical symptoms upon ingestion of HE. One child (P12) experienced vomiting and periorbital pruritus, P18 developed eczema, and P20 presented with vomiting. During the study period, P11 reintroduced HE based on low sIgE levels to ovomucoid (gal d1 < 0.10 kU/L) and decreasing sIgE to egg white (0.30 kU/L to 0.15 kU/L) and to ovalbumin (gal d2 0.31 kU/L to 0.11 kU/L). Highly heated HE (cake) and hard-boiled egg were successfully reintroduced. However, following the introduction of hard-boiled egg, EoE reactivated, necessitating the avoidance of all egg-containing products. P12 tolerated processed products containing HE, such as cookies and pancakes, at the time of inclusion, though lesser-heated egg products were not introduced during the follow-up period. P20 also started the introduction of HE based on low sIgE levels of ovomucoid (decreasing IgE of 0.37 kU/L to 0.27 kU/L) and decreasing sIgE to egg white (0.38 kU/L to 0.32 kU/L). Both highly heated and hard-boiled HE were introduced without immediate reactions, but the patient subsequently experienced EoE recurrence after consuming pancake and omelet, leading to renewed avoidance of all egg forms. P18 and P25 maintained a strict egg-free diet throughout the study, based on their high sIgE to egg white and its components and to egg yolk. In contrast, although P24 and P26 exhibited low sIgE levels to egg white, they preferred not to reintroduce HE and continued an egg-free diet.

#### 3.2.3. IgE Sensitization to Wheat

Among the thirty children in our study, nine children had detectable IgE levels to wheat (Figure 1, Table 3). Five children (P5, P10, P20, P21, and P24) exhibited concomitant IgE-mediated allergy to grass pollen, suggesting potential cross-reactivity to wheat [12]. Clinical reactivity was only observed in one child: P20 experienced abdominal pain following wheat ingestion, despite low specific IgE levels and concurrent grass pollen allergy. Other patients were on a diet free of wheat based on mildly elevated IgE, although they did not clinically react after wheat consumption. During the study period, P21 underwent a successful OFC to wheat, following a decrease in specific IgE to wheat from 2.16 kU/L to 0.61 kU/L, with no subsequent reactivation of EoE on gastroscopy.

#### 3.2.4. IgE Sensitization to Soy

In our cohort, elevated IgE levels to soy were detected in 5 out of 30 children (Figure 1, Table 4). Three of these children (P10, P11, and P21) had a confirmed birch pollen allergy, suggesting potential cross-reactivity with soy (cf. IgE to gly m4 is 100 kU/L in P21) [13]. Clinical reactivity was observed in one child (P20), who experienced vomiting following soy ingestion. Throughout the follow-up period, all children adhered to a strict soy-free diet.

#### 3.2.5. IgE Sensitization to Peanut, Tree Nuts, and Legumes

In our study cohort, 14 children had elevated IgE levels to peanut and/or tree nuts (Figure 1, Table 5 and Table 6). Clinical reactivity was observed in six cases: P4 developed a skin rash following tree nut ingestion, and P18 developed eczema following the ingestion of hazelnut. P12 exhibited dyspnea, angioedema, and urticaria after ingesting chocolate containing hazelnuts. P21 presented with symptoms of oral allergy syndrome (OAS) to hazelnut and walnut. Similarly, P10 experienced oral pruritus after consuming hazelnuts, and P23 experienced symptoms after the ingestion of peanuts consistent with OAS. During the observation period, P12 underwent OFCs with peanut, cashew, and almond, based on low sIgE levels and minimal skin prick test (SPT) responses. They were initially well tolerated, with no immediate clinical reactions. However, follow-up endoscopy revealed reactivation of EoE, leading to the reimplementation of dietary restrictions on peanut and all tree nuts. The remaining children maintained a peanut- and/or tree nut-free diet throughout the study period.

In literature on EoE, peanuts, lentils, beans, peas, chickpeas, and lupins are frequently categorized under the collective term ‘legumes’. As a result, legumes are often considered in association with peanuts within the context of food elimination therapy [6]. Therefore, in our study, we specifically evaluated sensitization to at least one of these legumes—excluding peanut, which was discussed separately above—in our pediatric EoE cohort. Four children adhered to a legume-free diet. P11 and P28 exhibited both positive sIgE and SPT results to legumes, whereas P18 and P24 showed elevated sIgE levels to legumes, although SPT results were not available for evaluation (Table 7).

#### 3.2.6. IgE Sensitization to Fish and Shellfish

In our cohort, six children had detectable IgE levels to fish and/or shellfish (Figure 1, Table 8). Clinical reactions to (shell)fish ingestion were observed in three children: P01 developed conjunctivitis and vomiting after consuming mussels, P11 experienced a skin rash after the ingestion of tuna (however, no IgE levels or SPT results to tuna were available), and P18 presented with eczema following white fish ingestion. None of the allergic children introduced fish or shellfish into their diet during the follow-up period.

#### 3.2.7. IgE Sensitization to Other Foods

Six out of thirty children avoided banana in their daily diet. P6 had a sIgE level of 0.12 kU/L to banana and a positive SPT. P10 experienced angioedema following banana ingestion and had an sIgE level of 0.33 kU/L. P11 developed vomiting after consuming bananas and had a positive SPT. P18 exhibited no clinical symptoms but had an sIgE titer of 2.46 kU/L to banana. P23 presented with symptoms consistent with oral allergy syndrome upon banana ingestion and avoided banana for this reason. P24 had a mildly elevated sIgE level to banana (0.14 kU/L).

Five children adhered to a sesame-free diet. P10 and P21 had a positive SPT for sesame, while P12 had an equivocal SPT result. P18 had a highly elevated sIgE level to Ses i1 (20.30 kU/L). P11 followed a sesame-free diet, though no clear documentation was available regarding the reason for dietary avoidance.

Four children (P18, P23, P24, and P25) followed a diet excluding chicken meat. All of these children had elevated sIgE levels to chicken meat, ranging from 0.83 kU/L to 2.62 kU/L. P18, P24, and P25 were known to have concurrent IgE-mediated HE allergy, which may be indicative of “egg-bird syndrome” [14].

Three children adhered to a buckwheat-free diet. P10 had an sIgE level of 1.33 kU/L to buckwheat and a positive SPT. P11 and P24 exhibited elevated sIgE levels to buckwheat, measuring 2.58 kU/L and 7.44 kU/L, respectively.

### 3.3. Symptom Score and Quality of Life Questionnaires

To assess the impact of disease remission and dietary restriction on EoE symptom score and/or on QoL, the following three questionnaires were recorded and analyzed. At each standard-of-care visit, two quality of life questionnaires (child PedsQL and parent PedsQL) and a symptom score questionnaire (PEESS) were provided to each study participant and his/her parents for completion. For children who completed more than one questionnaire, the follow-up duration ranged from 12 to 34 months.

#### 3.3.1. Analysis of PEESS Questionnaires

The PEESS questionnaire is intended for children aged 8 years and older. At the first time point, 20 children met this age criterion, of whom 19 completed the questionnaire. During follow-up, these children continued to complete a PEESS questionnaire at each subsequent time point. In total, 39 questionnaires were completed over the course of the study. The distribution of completed questionnaires per child was as follows: eight children completed one questionnaire, five children completed two questionnaires, four children completed three questionnaires, one child completed four questionnaires, and one child completed five questionnaires.

#### PEESS Symptom Scores in Relation to Remission Status

Among all completed questionnaires (n = 39), symptom scores were compared between children who were in remission at the time of questionnaire completion and those who were not. For the eight children who did not undergo a gastroscopy during follow-up, remission status was assessed based on the findings of their most recent gastroscopy performed prior to the questionnaire. The median PEESS score was 17.50 in children in remission (n = 31) and 23.75 in those not in remission (n = 8) (*p* = 0.1934), suggesting a trend toward greater symptom burden in children without EoE remission, although this difference was not statistically significant.

When restricting the analysis to the first time point only (n = 19), the median PEESS score was 20.00 in the remission group (n = 12) and 27.50 in the non-remission group (n = 7) (*p* = 0.3293). While not reaching statistical significance, this again indicated a tendency toward higher symptom severity in children without histological remission.

#### PEESS Symptom Scores in Relation to Treatment Type

A similar analysis across all completed questionnaires (n = 39) compared children receiving dietary treatment (with or without pharmacological therapy) to those on pharmacological therapy alone. Median symptom scores were identical in both groups, at 20.00 (diet group: n = 26; non-diet group: n = 13; *p* = 0.9472). No significant differences were observed based on the number of foods eliminated, whether 0, 1–2, 3–4, 5–6, or more than 6 (*p* = 0.8490).

At the first time point only (n = 19), the median symptom score in the dietary group (n = 16) was 24.38, compared to 20.00 in the non-diet group (n = 3) (*p* > 0.9999). Again, no significant differences were observed between children on more extensive versus limited food elimination diets (*p* = 0.9513).

#### 3.3.2. Analysis of QoL Questionnaires

The child self-report version of the PedsQL is intended for children aged 5 years and older. At the first time point, 27 children met this criterion and completed the child PedsQL. These same children continued to complete the questionnaire at each subsequent time point during follow-up. Additionally, one child—who was almost 4 years old at the start of the study—reached the age of 5 by the fourth (and last) time point and completed the questionnaire for the first time at that visit. In total, 47 child self-report questionnaires were completed throughout the study period. The distribution of completed questionnaires per child was as follows: 13 children completed one questionnaire, 6 children completed two questionnaires, 6 children completed three questionnaires, and 1 child completed four questionnaires.

The parent proxy-report version of the PedsQL is available for children aged 2 years and older. At the first time point, all participating children met this age requirement. However, one parent failed to correctly complete the proxy-report form—rendering it invalid—while another did not complete the questionnaire. In total, 58 valid parent proxy-report questionnaires were collected during the study. The distribution was as follows: 12 parents completed one questionnaire, 6 parents completed two questionnaires, 7 parents completed three questionnaires, 2 parents completed four questionnaires, and 1 parent completed five questionnaires.

#### QoL in Relation to Remission Status

Among all completed questionnaires, there were no significant differences in QoL observed between children with EoE in remission and those not in remission. The median Child PedsQL score was 67.86 in the remission group (n = 38) and 65.48 in the non-remission group (n = 9) (*p* = 0.3134). Parent-reported QoL scores were also similar, with median scores of 69.44 for children in remission (n = 48) and 65.36 for those not in remission (n = 10) (*p* = 0.4717).

When again limiting the analysis to the first time point only, there were no significant differences in QoL observed between children with EoE in remission and those not in remission. The median Child PedsQL score was almost identical in the remission group in comparison to the non-remission group: 70.58 in the remission group (n = 19) and 70.08 in the non-remission group (n = 8) (*p* = 0.9285). Parent-reported QoL scores showed median scores of 77.46 for children in remission (n = 20) and 71.75 for those not in remission (n = 8) (*p* = 0.6718).

Seven children achieved remission during their follow-up period. QoL scores recorded during remission were compared to those obtained from the same child at a different time when they were not in remission (Table 9). No consistent or systematic differences in QoL scores were observed based on remission status, suggesting that QoL did not markedly decline during non-remission periods compared to remission periods. P01 (timepoint a) and P02 lost remission despite no changes in therapy. At the time of non-remission, P01 increased the dose of tCS to 2 mg twice daily (instead of once daily), while P02 initiated tCS in addition to PPI and a diet free of cow’s milk and (pea)nuts. Both patients subsequently regained remission. Patients P11, P21, P22, and P25 were not in remission at their initial evaluation. P11 adhered to a strict six-food elimination diet (FED), supplemented with potato, buckwheat, sesame, and lupin seed. P25 was receiving high-dose PPI and initiated tCS, whereas P21 and P22 started tCS alongside a 5FED. All patients subsequently achieved remission.

#### QoL in Relation to Treatment Type

In our cohort of thirty children, six received exclusively pharmacological treatment at baseline, comprising one child on tCS and five children treated with PPIs. Two children adhered to a strict dietary regimen, while twenty-two received a combination of pharmacological therapy and dietary management. Additionally, one child undergoing combined pharmacological therapy and dietary management required enteral nutrition via a nasogastric tube. In total, at 60 different timepoints, a child and/or parent PedsQL questionnaire was completed, and the dietary strategy at these visits is summarized in Table 10.

In the total sample of completed questionnaires, QoL assessment showed no significant differences in QoL between children with EoE following a dietary restriction in comparison to those without dietary restrictions. The QoL tended to be slightly better in the non-diet group in comparison to the diet group (the median Child PedsQL score of 65.70 in the diet group (n = 36) and 75.60 in the non-diet group (n = 11) (*p* = 0.2010)). Parent-reported QoL scores (Parent PedsQL) were slightly inverse, with median scores of 71.13 for the diet group (n = 41) and 64.29 for the no-diet group (n = 17) (*p* = 0.1463). However, a significant difference in QoL was observed across groups with varying degrees of dietary restriction, categorized by the number of eliminated foods (0, 1–2, 3–4, 5–6, and >6) in both child- and parent-reported assessments: Child PedsQL scores differed significantly between groups (*p* = 0.0102, n = 47), as did the Parent PedsQL scores (*p* = 0.0203, n = 58). Additional Dunn’s post hoc analysis showed a higher QoL in children who were on limited food restriction (one or two FED) than in children who were not on a diet or in children with more than two food restrictions (Table 11).

When focusing solely on the first time point, QoL assessment showed no significant differences in QoL between children with EoE following a dietary restriction in comparison to those without dietary restrictions with a median Child PedsQL score of 70.58 in the diet group (n = 21) and 67.75 in the non-diet group (n = 6) (*p* = 0.7628). Parent-reported QoL scores were similar, with median scores of 73.67 for the diet group (n = 21) and 71.42 for the non-diet group (n = 7) (*p* = 0.8765).

Additionally, no significant difference in QoL was observed across groups with varying degrees of dietary restriction (0, 1–2, 3–4, 5–6, and >6 foods eliminated) in both child- and parent-reported assessments: *p* = 0.5243 in child PedsQL (n = 27) and *p* = 0.2306 in parent PedsQL (n = 28). Similarly to the total group, post hoc pairwise comparison observed a trend towards better QoL scoring in children on a one or two-food elimination diet than in children without a diet or in children who are on more than two food restrictions. Although this trend was observed, it no longer reached statistical significance (Table 11).

## 4. Discussion

This study provides novel insights into the difficult overlap between food-specific IgE sensitization/allergy and eosinophilic inflammation in children with EoE. Our findings suggest that IgE sensitization is prevalent among pediatric EoE patients and may have clinical implications for EoE disease management and patient well-being.

A key observation in our study was the high prevalence of IgE sensitization to food allergens in children with EoE. This finding aligns with prior research indicating that EoE is frequently associated with allergic comorbidities in children with EoE, including IgE-mediated food allergies and environmental allergies [1]. While the high prevalence of IgE sensitization is not a novel finding in itself, documenting these data within a real-world, single-center pediatric cohort in Belgium adds value by contributing geographic and clinical context. Regional variations in dietary habits, allergen exposure, and clinical management practices can influence sensitization patterns. Therefore, the documentation of these local data may contribute to a more nuanced understanding across populations. The mechanisms underlying this relationship remain incompletely understood, yet it is hypothesized that epithelial barrier dysfunction and immune dysregulation contribute to both EoE and IgE sensitization [15,16]. In contrast, IgE-mediated food allergy appears to be less commonly described in adult patients with EoE [17]. Despite the high prevalence of IgE sensitization to food allergens reported in children with EoE in the literature, clinically manifest allergic reactions such as anaphylaxis are rare [18]. In our cohort, most reactions were limited to mild cutaneous or gastrointestinal symptoms, and no cases of severe reactions or anaphylaxis were observed. As a result, the presence of sensitization has to be considered in clinical decision-making to guide dietary management, particularly when supported by a suggestive clinical history or diagnostic work-up.

Given this overlap between EoE and IgE-mediated allergies, distinguishing between IgE-mediated and non-IgE-mediated symptoms poses a clinical challenge, as overlapping manifestations—such as vomiting and abdominal pain—can mimic one another. This differentiation becomes even more difficult in cases of mild IgE sensitization with subtle symptoms. Nevertheless, IgE-mediated food allergy symptoms typically occur rapidly—within two hours after ingestion of the culprit food—whereas EoE is characterized by a delayed onset of symptoms [19]. The diagnostic complexity highlights the need for a comprehensive evaluation of both IgE-mediated allergic mechanisms and EoE-specific pathophysiology in pediatric patients.

This difficult overlap between food-specific IgE sensitization/allergy and EoE extends to dietary management, which is particularly complex in patients with EoE who also present with IgE-mediated food allergies. Moreover, whereas dietary management for EoE is typically maintained on a continuous basis, dietary avoidance in IgE-mediated food allergy may be temporary and subject to re-evaluation over time [20,21]. Indeed, some children eventually outgrow their IgE-mediated food allergy. The development of tolerance, however, follows a variable natural course that depends on the specific food allergen involved. In IgE-mediated CM allergy, the resolution rates have been reported as 19% by four years of age in the United States and 42% by eight years, 64% by the age of twelve years, and 79% by sixteen years of age. In a Korean cohort, 50% of children outgrew their CM allergy at a median age of 8.7 years [21]. Similarly, for HE allergy, the literature demonstrates that 73–89% of children with an IgE-mediated HE allergy outgrow their allergy by six years of age [21]. In IgE-mediated wheat allergy, 27% of children in a Thai cohort achieved tolerance by four years of age and 69% by nine years. In a Polish cohort, wheat allergy resolution was observed in 20% of children by four years of age, 52% by eight years, 66% by twelve years, and 76% by eighteen years [21]. Resolution rates for IgE-mediated soy allergy have not been extensively studied. In a U.S. cohort, tolerance development was observed in 25% of children by four years of age, 45% by six years, and 69% by ten years [21]. The natural course of tree nut allergy in children remains poorly studied. A 2005 publication reported that 8.9% of children with prior clinical reactivity and sensitization to tree nuts outgrew their allergy. In contrast, resolution rates for IgE-mediated peanut allergy are well documented, with studies indicating that 22% of children and adolescents outgrow peanut allergy by four years of age and 29% by six years [21]. Natural tolerance development for fish and shellfish is less common compared to other food allergens. A Greek study reported that complete tolerance to fish increased with age, from 3.4% in preschool-aged children to 45% in adolescents. Similarly, a study in Canada estimated resolution rates of 0.6% per person-year for fish and 0.8% per person-year for shellfish [21]. Mostly, the reintroduction of a specific allergen in IgE-mediated food allergy can be guided by food-specific IgE decline [22]. However, a decline in IgE levels to a specific food over time does not necessarily indicate that the food can be safely reintroduced, and the presence of specific IgE levels does not necessarily rule out tolerance [23]. While the absence of immediate IgE-mediated symptoms after ingestion may suggest clinical tolerance, the potential for EoE reactivation remains, necessitating renewed and, later on, continued dietary restriction. These considerations underscore the importance of individualized dietary strategies that address both IgE-mediated and non-IgE-mediated mechanisms of disease.

Besides the decline in specific IgE, a decline in food-specific IgE components might point to partial tolerance induction. Indeed, as observed in IgE-mediated CM or HE allergy, tolerance can develop gradually over time through exposure to progressively less heated forms of the allergen [24]. The introduction of highly heated proteins can even accelerate the process of tolerance induction [25]. Potentially, heated allergens could also play a role in food tolerance induction in EoE. In a recent case report, we described two children with EoE (also included in this cohort, P11 and P20) in whom the introduction of extensively heated HE was well tolerated, whereas subsequent exposure to less heated forms of HE (hard-boiled egg and pancake, respectively) resulted in disease reactivation [26]. These findings suggest that certain heated proteins may be tolerated in some children with EoE. To further investigate whether this approach could safely broaden dietary options, we are currently conducting the PedEoE IgE study (NCT06381219), which explores the feasibility of introducing extensively heated CM or HE in a cohort of children with CM- or HE-induced EoE, while remaining in remission. This might have a positive impact on the children’s QoL.

Indeed, beyond dietary considerations, the impact of multiple food restrictions on QoL in pediatric EoE patients is another critical aspect of EoE management. In our study, we observed that dietary restrictions negatively affected children’s reported QoL and that there is a significant difference in QoL between groups with varying degrees of dietary restriction. Children avoiding one or two food products seem to have better QoL scores (child PedsQL as well as parent PedsQL scores) than those who have to avoid more than two food products and/or those who are on pharmacological treatment only. While our findings suggest better QoL in children following a 1–2 food elimination diet, it is also possible that this observation partially reflects a less severe or less complex disease phenotype in this subgroup. As our study was not powered to adjust for all potential confounding variables, such as disease severity or allergic sensitization complexity, this association should be interpreted in light of the study’s exploratory nature and limited sample size. Despite the limitations imposed by a relatively small sample size, our cohort represents one of the largest single-center pediatric EoE datasets in the Flanders region. Given that eosinophilic esophagitis is considered a rare disease, particularly in children [3], collecting data from 30 pediatric patients in a single tertiary referral center would still offer valuable insights. Adult patients were deliberately excluded from this study to preserve the homogeneity of the cohort. Pediatric and adult EoE populations differ in disease phenotype, symptomatology, and treatment response, and combining these groups could have introduced additional variability, particularly in the assessment of quality of life, which was conducted using tools specifically validated for pediatric populations. Furthermore, a multicenter design across different regions or countries was not pursued in order to avoid heterogeneity in dietary patterns and feeding practices, which are known to vary significantly by geography and culture. Since dietary management is a central focus of this study, maintaining a consistent regional context was essential to ensure internal validity and interpretability of the findings.

Also, given the observational and non-randomized nature of this study, we cannot exclude the possibility that confounding factors such as disease severity, allergic profile complexity, and parental preferences influenced both treatment choice and QoL outcomes. While this may limit causal interpretation, the real-life setting of the study allows for the inclusion of a diverse patient population that more closely resembles everyday clinical practice. When interpreting the QoL and symptom data, it should be noted that analyses were performed using both all available completed questionnaires per child and only the first completed questionnaire per patient, in order to reduce potential bias introduced by repeated measurements. Although the inclusion of multiple questionnaires from the same individual across follow-up time points may introduce within-subject dependency, this approach was deemed appropriate to capture temporal patterns within a real-world pediatric EoE population. Given the limited sample size and observational design, non-parametric statistical tests were applied. Nonetheless, the use of more advanced statistical techniques, such as mixed-effects models, may be more suitable in future studies with larger cohorts to formally account for repeated measures and potential confounding variables.

Lastly, the accuracy and reliability of questionnaire-based QoL assessments in pediatric EoE patients must be carefully considered. Parental influence during questionnaire completion may introduce bias, as parents may inadvertently guide their child’s responses when present in the room. Additionally, young children may struggle to recall past events accurately, as their sense of time is still developing. These factors should be considered when interpreting QoL data in a young study population.

## 5. Conclusions

Although based on a relatively small, convenience-based sample, our findings suggest that IgE sensitization may play a meaningful role in shaping the clinical presentation and quality of life in children with EoE. Additionally, the incorporation of multidisciplinary care approaches may enhance disease management and improve overall patient outcomes. Further research in larger and more diverse pediatric populations is needed to clarify the role of IgE sensitization in EoE pathophysiology and to develop tailored interventions that address both the IgE- and non-IgE-mediated aspects of the disease. In this age group, the concurrent development of tolerance to specific food allergens in IgE-mediated food allergy may present an additional challenge for pediatricians managing EoE in children compared to EoE in adolescents or adults.

## Figures and Tables

**Figure 1 nutrients-17-01980-f001:**
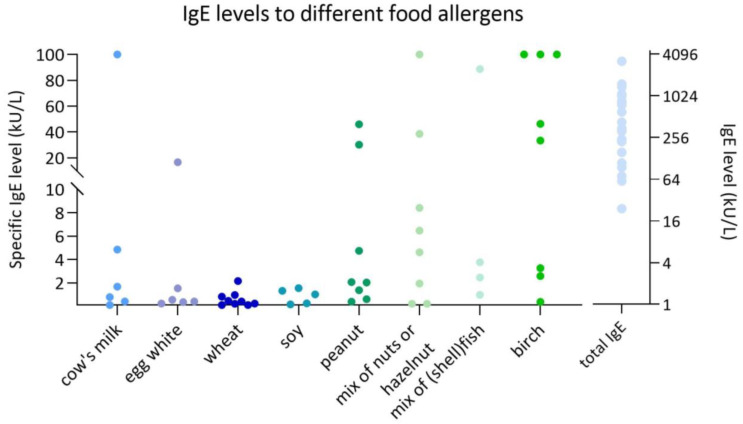
Total IgE level (kU/L) and specific IgE levels (kU/L) to different food allergens in individual study patients.

**Table 1 nutrients-17-01980-t001:** IgE levels (kU/L) to cow’s milk and its components in individual study patients at inclusion ± 5 months.

ID	Total IgE	IgE Cow’s Milk	IgE Bos d4	IgE Bos d5	IgE Bos d6	IgE Bos d8
P02	596	100.00	-	-	-	100.00
P07	3216	1.68	-	-	-	-
P09	225	0.10	-	-	-	-
P11	60	-	0.24	0.22		0.10
P14	24	4.86	-	-	-	-
P24	348	0.42	-	-	-	-
P26	94	0.78	-	-	-	-

**Table 2 nutrients-17-01980-t002:** IgE levels (kU/L) to hen’s egg and its components in individual study patients at inclusion ± 5 months.

ID	Total IgE	IgE Egg White	IgE Gal d1	IgE Gal d2	IgE Gal d3	IgE Egg Yolk
P11	60	-	0.10	0.11	0.10	-
P12	156	0.42	0.10	-	-	0.18
P18	1022	16.60	4.93	6.97	4.40	15.70
P20	110	0.36	0.10	-	-	-
P24	348	0.23	-	-	-	-
P25	860	1.54	0.46	0.25	-	14.70
P26	94	0.55	-	-	-	-

**Table 3 nutrients-17-01980-t003:** IgE levels (kU/L) to wheat and its components in individual study patients at inclusion ± 5 months.

ID	Total IgE	IgE Wheat	IgE Tri a14	IgE Tri a19
P03	793	0.47	-	-
P05	242	0.96	0.10	-
P07	3216	0.24	-	-
P09	225	0.10	-	-
P10	314	0.83	0.10	0.10
P20	110	0.11	-	-
P21	1377	2.16	-	-
P24	348	0.41	-	-
P26	94	0.22	-	-

**Table 4 nutrients-17-01980-t004:** IgE levels (kU/L) to soy and its components in individual study patients at inclusion ± 5 months.

ID	Total IgE	IgE Soy	IgE Gly m4	IgE Gly m5	IgE Gly m6
P10	314	1.33	-	-	-
P11	60	1.56	-	0.10	0.10
P20	110	0.17	0.10	0.10	0.11
P21	1377	1.02	100.00	-	-
P24	348	0.25	-	0.27	-

**Table 5 nutrients-17-01980-t005:** IgE levels (kU/L) to peanut and its components in individual study patients at inclusion ± 5 months.

ID	Total IgE	IgE Peanut	IgE Ara h1	IgE Ara h2	IgE Ara h3	IgE Ara h6	IgE Ara h8	IgE Ara h9
P05	242	0.60	0.10	0.10	0.10	-	0.10	0.10
P08	751	2.03	-	-	-	-	-	-
P10	314	1.37	0.10	0.10	0.10	-	-	-
P11	60	4.74	0.10	0.10	0.10	0.10	8.68	0.12
P12	156	-	-	0.15	-	0.10	-	-
P16	1511	2.08	-	-	-	-	-	-
P22	427	46.00	1.51	32.1	0.34	-	5.63	0.10
P23	1086	30.10	0.69	0.10	0.15	-	100.00	0.35
P24	348	0.39	0.10	0.10	-	-	-	-

**Table 6 nutrients-17-01980-t006:** IgE levels (kU/L) to the different tree nuts and their components in individual study patients at inclusion ± 5 months.

ID	Total IgE	IgE Mix of Nuts	IgE Hazelnut	IgE Cor a1	IgE Cor a8	IgE Cor a9	IgE Cor a14	IgE Cashew	IgE Ana o3	IgE Pistachio	IgE Walnut	IgE Almond
P04	327	-	4.62	-	0.10	0.10	0.10	-	-	-	-	-
P05	242	-	-	0.18	0.10	0.10	0.10	0.10	-	-	-	-
P06	71	0.22	-	-	-	-	-	-	-	-	-	-
P10	314	-	8.42	-	-	0.10	0.10	-	-	-	-	-
P11	60	-	38.5	-	-	0.16	0.10	0.19	0.11	2.93	2.34	1.53
P12	156	-	6.47	0.10	0.10	12.20	8.41	0.26	0.46	0.27	-	-
P18	1022	-	-	-	-	18.00	26.60	-	-	-	-	-
P20	110	0.21	-	-	-	-	-	-	-	-	-	-
P21	1377	100.00	-	-	-	-	-	-	-	-	-	-
P22	427	-	1.94	-	-	0.10	0.10	-	-	0.10	-	0.16

**Table 7 nutrients-17-01980-t007:** IgE levels (kU/L) of the different legumes in individual study patients at inclusion ± 5 months.

ID	Total IgE	IgE Peas	IgE Chickenpeas	IgE Lentils	IgE Beans	IgE Lupins
P11	60	3.17	3.11	2.88	4.02	2.74
P18	1022	11.80	8.94	5.02	0.88	-
P24	348	2.36	-	-	0.52	-
P28	87	1.92	-	2.44	0.85	-

**Table 8 nutrients-17-01980-t008:** IgE levels (kU/L) to fish and shellfish in individual study patients at inclusion ± 5 months.

ID	Total IgE	IgE (Shell)Fish Mix	IgE Mussel	IgE Pen a1	IgE Cod	IgE Gad c1
P01	222	88.70	24.80	-	-	-
P11	60	-	0.18	0.10	-	0.10
P16	1511	0.98	-	-	-	-
P18	1022	-	-	-	9.94	8.04
P23	1086	2.47	-	0.10	-	3.59
P24	348	3.77	-	-	-	-

**Table 9 nutrients-17-01980-t009:** PEESS and PedsQL scores of children who changed remission status during the follow-up period. The arrow presented in the table denotes the direction of the change in remission status. NA = not applicable.

ID	Remission				No Remission		
	PEESS	Child PedsQL	Parent PedsQL		PEESS	Child PedsQL	Parent PedsQL
P01-a	12.50	79.00	79.83	→	20.00	75.83	93.33
P01-b	17.10	77.00	83.33	←	20.00	75.83	93.33
P02	NA	NA	38.54	→	NA	NA	31.25
P11	NA	43.33	66.25	←	NA	72.50	85.00
P21	17.50	70.17	78.92	←	36.75	75.17	64.42
P22	26.25	82.50	74.75	←	15.00	77.00	89.25
P25	10.00	89.92	80.58	←	6.25	90.83	63.32

**Table 10 nutrients-17-01980-t010:** Type of food elimination diet (FED) at time of questionnaire.

	0 FED	1–2 FED	3–4 FED	5–6 FED	>6 FED
n	12	25	4	7	12

**Table 11 nutrients-17-01980-t011:** QoL assessment in relation to treatment type, using the Kruskal–Wallis test followed by post hoc Dunn’s test; significant values are indicated with *.

Number of Completed Questionnaires		Type of PedsQL	Type of FED	Mean	Kruskal Wallis Test	Dunn’s Post Hoc Test		
						0 FED vs. 1–2 FED	0 FED vs. >2 FED	1–2 FED vs. >2 FED
All questionnaires	n = 47	Child PedsQL	0 FED	61.97	*p* = 0.0251 *	*p* = 0.4640	*p* > 0.9999	*p* = 0.0235 *
			1–2 FED	74.35				
			>2 FED	61.19				
	n = 58	Parent PedsQL	0 FED	58.44	*p* = 0.0070 *	*p* = 0.0060 *	*p* = 0.3601	*p* = 0.2336
			1–2 FED	73.40				
			>2 FED	65.92				
First time point questionnaires	n = 27	Child PedsQL	0 FED	64.93	*p* = 0.4155	*p* > 0.9999	*p* > 0.9999	*p* = 0.5954
			1–2 FED	73.66				
			>2 FED	66.06				
	n = 28	Parent PedsQL	0 FED	72.05	*p* = 0.0614	*p* = 0.6251	*p* > 0.9999	*p* = 0.0556
			1–2 FED	82.00				
			>2 FED	64.68				

## Data Availability

The raw data supporting the conclusions of this article will be made available by the authors on request.

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
