# Peer review of "IgE-Mediated Food Sensitization, Management Strategies, and Quality of Life in Pediatric Eosinophilic Esophagitis: A Prospective Observational Study"

_nutrients, 2025, doi:10.3390/nu17121980_

Round 1

Reviewer 1 Report

Comments and Suggestions for Authors

This prospective observational study investigates the prevalence of IgE-mediated food sensitization, dietary management strategies, and their impact on Quality of Life (QoL) in 30 children with eosinophilic esophagitis (EoE) at a single tertiary center in Belgium. The authors collected data on demographics, allergy testing, dietary history, treatment, and QoL/symptom scores (PedsQL/PEESS) over a median follow-up of 14.5 months. They report a high prevalence of IgE sensitization to common food triggers and find that QoL scores differed significantly based on the degree of dietary restriction, with better QoL reported for children and parents when 1 or 2 food products were excluded compared to more extensive restrictions or medication-only treatment. While the topic of co-existing IgE sensitization and EoE in children, and its impact on management and QoL, is clinically relevant and important, this manuscript, in its current form, has several critical limitations that preclude its publication in Nutrients. The primary concerns relate to the small sample size which severely limits statistical power and the generalizability of the findings, particularly for the QoL analyses. Additionally, the interpretation of some results is problematic, and the conclusions drawn are often not adequately supported by the presented data. The study design, while prospective, also has inherent limitations that are not sufficiently addressed.

Major concerns:

  1. The most significant flaw is the small sample size (N=30 PedEoE patients) for drawing robust conclusions, especially regarding the QoL analysis which is a central aim of the study. The analysis of PEESS scores in relation to remission status (Section 3.3.1.1) and treatment type (Section 3.3.1.1 - this section number appears to be a typo and should likely be 3.3.1.2) also suffers from small subgroup sizes (e.g., n=8 for non-remission in PEESS analysis across all time points; n=3 for non-diet group at first time point). The authors acknowledge non-significance but suggest trends; however, with such low power, even observing a trend is speculative. In addition, the study aims to assess the impact of dietary regimens compared to pharmacological treatments on QoL (Lines 83-85). However, the "medication only" group (0 foods eliminated) is very small (n=11 for Child PedsQL, n=17 for Parent PedsQL across all time points, see Table 10), making robust comparisons difficult.
  2. The core finding regarding QoL and dietary restriction (better QoL with 1-2 food eliminations) is counterintuitive if taken at face value without deeper consideration of confounding factors. It is plausible that children with less severe or less complex EoE (or fewer concomitant IgE allergies) are the ones for whom a 1-2 food elimination diet is sufficient and effective. Their inherently better disease state or simpler management could be the primary driver of better QoL, rather than the number of foods eliminated being the direct cause of better QoL. The study does not adequately control for or discuss disease severity, complexity of allergic sensitization profiles, or the effectiveness of the specific dietary intervention in achieving remission when interpreting these QoL differences. Also, the authors state "QoL scores differed significantly (child p = 0.0102; parent p = 0.0203), with better QoL for children and parents when 1 or 2 food products were excluded from the diet in comparison to more excluded food products or treatment by medication only" (Abstract, Lines 32-35). However, Table 11 (all time points) shows that for Child PedsQL, the 1-2 FED group is significantly better than the 0 FED group, but not significantly different from the 3-4, 5-6, or >6 FED groups. For Parent PedsQL, the 1-2 FED group is significantly better than the 0 FED and >6 FED groups, but not the 3-4 or 5-6 FED groups. The conclusion in the abstract oversimplifies these nuanced (and still underpowered) findings.
  3. The observational nature of the study means treatment decisions were not randomized. Factors influencing treatment choice (e.g., severity of EoE, number/type of IgE sensitizations, parental preference, previous treatment failures) are likely confounders that could influence both the type of management and the QoL outcomes. The manuscript does not adequately address or adjust for these potential confounders in the analysis of QoL and treatment type.
  4. The number of gastroscopies per child varied widely (0 to 7), meaning remission status was not assessed uniformly. For the 8 children with no gastroscopies during follow-up, how was remission status determined for the QoL/PEESS analysis at those time points The number of completed questionnaires per child/parent also varied (1 to 5). Analyzing "all completed questionnaires" pools data from different time points and different numbers of contributions per patient, which can introduce bias if not handled with appropriate statistical methods (e.g., mixed-effects models, which were not used). The current analysis treats these as independent observations, which they are not.
  5. While the study prospectively documents IgE sensitization, the finding that IgE sensitization is prevalent in pediatric EoE (Section 3.2, Discussion Lines 507-509) is well-established in the literature, as acknowledged by the authors. The specific prevalence figures from this small, single-center Belgian cohort, while adding to the geographical data, do not represent a major novel contribution on their own. The clinical vignettes are illustrative but do not form a systematic analysis of management challenges beyond anecdotal evidence.
  6. Section 3.2 extensively details sIgE levels for individual patients for various foods. While this provides a characterization of the cohort, it is largely descriptive. The study does not systematically correlate the levels of sIgE or the number of sensitizations with EoE severity, treatment response (beyond individual cases), or QoL in a rigorous statistical manner. The narrative focuses on individual patient journeys with reintroductions and EoE reactivation, which, while clinically interesting, does not lead to generalizable conclusions due to the small numbers and lack of a control group for these specific interventions.

Minor concerns:

  1. The manuscript uses "IgE-mediated food sensitization" and "IgE-mediated food allergy" somewhat interchangeably. While sensitization is a prerequisite for allergy, they are not the same. Greater precision in terminology would be beneficial, especially when discussing clinical reactivity.
  2. The rationale for the specific cut-off of "> 0.10 kU/L" for sensitization (Line 123) should be briefly justified, as different cut-offs are sometimes used.
  3. The discussion on the natural course of IgE-mediated food allergies (Lines 532-556) is extensive and provides good background but feels somewhat disconnected from the study's own findings, as the study did not systematically track tolerance development.
  4. The statement "QoL assessments were conducted using Mann-Whitney U test or the Kruskal-Wallis test with the Dunn’s test as post-hoc test" (Lines 120-121) is a bit general. Specify when each was used (e.g., Mann-Whitney for two-group comparisons, Kruskal-Wallis for >2 groups).

Author Response

Dear reviewer, thank you for your comments and questions.
Please, see the attachment for a point-by-point reply.

Kind regards,
Lisa Nuyttens

Reviewer 2 Report

Comments and Suggestions for Authors

The article presents data of the prospective observational study on IgE-mediated food sensitization, management strategies and quality of life in pediatric eosinophilic esophagitis. Dta are interesting, results are presented extensively and conclusions clearly written. Before considering the article for publications I suggest some corrections and also to clarify some questions not explained in the text.

  1. Modes of therapy in pediatric EoE are presented in the Introduction. However, modes of therapy in study patients were not explained probably because of the complexity and individual approach. Nevertheless I recommend that the authors explain modes of treatment in their patients in the text.
  2. Line 81: in the entire paragraph sentences are too long, also word order in sentences should be changed according to English language rules.
  3. Line 93: instead "diagnosis was retained" I suggest the expression "diagnosis was established" or "made".
  4. Line 96: I suggest to write that the exclusion criteria were "age above 16 years..." and "other conditions that cause esophageal eosinophilia (gastroesophageal reflux disease or some other if they were presented, such as Crohn's disease). 
  5. Line 137: since cystic fibrosis is a lifelong genetic disease it is correct to write "Patient 12 was diagnosed with cystic fibrosis" instead "had a history of CF".
  6.  Line 138: since EoE is well-known for multiple endoscopies during follow-up I suggest that the authors explain the reason why 8 patients had no endoscopies during follow-up.
  7. I suggest that the abbreviation for a word "patient" (P) stands after first mentioning the entire word.
  8. I suggest to write the entire words in headlines of tables (cow's milk, hen's egg), not the abbreviations. Also, as written in the text, it is better to write "IgE to...) instead of "IgE of..." in all headlines of tables.
  9. I suggest that tables 4, 7, 8 and further should be separated from text.

Author Response

Dear reviewer,

Thank you for your comments and questions.

Please, see the attachment for a point-by-point reply.

Kind regards,

Lisa Nuyttens

Reviewer 3 Report

Comments and Suggestions for Authors
  • The chapter on „study population“ includes also aspects of data collection. The data collection needs to be described in more detail.
  • Has any sample size calculation been conducted a priori?
  • Has there been any loss-to-follow-up?
  • The sample seems to be much too small to conduct further analyses.

Author Response

(The authors gave the same response as above.)

Reviewer 4 Report

Comments and Suggestions for Authors

The paper dedicated  for reviewing procedure  is an interesting and well-prepared original article presenting the results of laboratory tests in eosinophilic esophagitis of the pediatric population – a group of 30 children with IgE-dependent hypersensitivity to food. Additional survey studies of the quality of life of children and their parents, on the day of admission during desensitization and during the desensitization period and after food challenge additionally enrich the analysis of the problem of food allergies. The authors of the article used the appropriate statistical analysis of the developed research results. Although the analysis of the research results presents only trends usually without statistical significance with a small study group, each of such interesting data is important for the overall picture of the entire presented problem. An interesting discussion presents the picture of the problem in the world and also factors influencing the obtained research results.

Author Response

Dear reviewer,

We would like to thank you for this feedback and evaluation of our manuscript.

Kind regards,

Lisa Nuyttens

Round 2

Reviewer 1 Report

Comments and Suggestions for Authors

Can be accepted.

Reviewer 3 Report

Comments and Suggestions for Authors

My previous concerns regarding the methodological issues (small sample size) still apply. For that reason, I leave the decision to the editor.